# A Study on Double Inputs Direct Contact and Single Output Capacitively Coupled Conductivity Detector

**DOI:** 10.3390/s22072729

**Published:** 2022-04-02

**Authors:** Shuangfei Zhang, Hongyan Yuan, Dan Xiao

**Affiliations:** College of Chemical Engineering, Sichuan University, Chengdu 610017, China; zsfzsy2020@163.com (S.Z.); yuan_hy@scu.edu.cn (H.Y.)

**Keywords:** capacitive coupling, conductivity detection, contact, contactless, impedance

## Abstract

In this paper, an improved double inputs direct contact and single output capacitively coupled conductivity detector (DISODCD) based on traditional contactless capacitively coupled conductivity detector (C^4^D) is developed. The sensor uses double inputs of the contact electrode and capacitively coupled output of the contactless electrode and a lock-in amplifier to reduce interfering noise signals and amplify gain. Parallel circuit counteracts the part of the adverse capacitance reactance introduced by electrode polarization and reduces the effect of the impedance caused by the coupled wall capacitance to measure the resistance of solution. The sensor reduces limit of detection (LOD) of analyte and improves the sensitivity of the device. The LOD of the potassium chloride solution is 1 nM, and the detection range is 0.01 μM to 10 mM in actual testing for a single sample. The ratio of the response of potassium chloride solution to background ultrapure water at low concentrations is better than that of double input capacitively coupled contactless conductivity detector (DIC^4^D) and direct contact conductivity detection (DCD) under the same condition. In the case that the test cell is contaminated with impurities, pollution of impurities has little effect on the response of DISODCD. In practical application, it has a good service life.

## 1. Introduction

The conductivity detection method is used widely in components analysis technology and is mainly suitable for the conductivity detection and analysis of samples separated by capillary electrophoresis. In conventional conductivity detection techniques, the electrodes are in direct or non-contact with fluid medium or electrolyte solution. Contact detection is simple to build up and can be used by multiple conductance detectors simultaneously, but it is easy to contaminate electrodes or electrochemical attack electrodes. The instrument is expensive because the capillary tube requires some modifications [1,2]. The non-contact method solves the problems such as electrode contamination and separation of high voltage interference, but the detection limit is relatively low.

Conductivity measurements were made using contact electrodes early [3,4,5]. Thereafter, Huang et al. [6,7,8,9] designed an on-column conductivity detector for capillary zone electrophoresis and made a series of improvements and combinations. In 1980, Gaš et al. [10] proposed a high-frequency contactless conductivity detector for a new detection system for isotachophoresis. Since then, many researchers have explored conditions and improved methods on this basis [11,12,13,14,15]. Capacitive reactance elimination principle [16,17,18,19,20,21,22,23] provides another idea for non-contact conductivity detection, but the performance improvements are very limited. These improvements in detection performance are relatively singular, and there is little research on the principles.

In recent years, improvements and innovations about contact conductivity detectors are few. Research on method of the conductivity detection for improving performance is listed in Appendix A [24,25,26], high-performance liquid chromatography (HPLC), ultraviolet detector (UV), di. Most current research focuses on the combination of conductivity detector and other detection instruments. It includes capillary electrophoresis (CE) [27,28,29,30], high-performance liquid chromatography (HPLC), ultraviolet detector (UV), diode array detector (DAD), fluorescence detector (FLD), and other forms of combination [31,32,33] so that the separation of samples and detection methods can achieve practical combination. There are few reports on improving the sensitivity of conductivity and the limit of detection of solution concentration in the conductivity detection device. In basic research on the performance of sensors, common chemicals are usually used, such as potassium chloride solution used in our work that is ionized fully at both low and high concentrations, which provides the exact results and verification of a new approach. Although the DCD has good response sensitivity, it is eliminated due to electrode polarization (EP) and other problems.

In the actual detection, the electrodes of most conductivity detectors are in direct contact with the solution, and the performance of sensitivity and resolution is excellent. However, electrode polarization occurs in the use of the electrode, which causes the measured value of conductivity to deviate from the true value. Meanwhile, the chemical reaction may occur between the solution to be detected and the electrode or the solution adhering to the electrode, resulting in contamination of the electrode during long-term use. The capacitively coupled contactless conductivity detector can avoid problems of polarization and contamination of the electrode, however, it pays the price of reduced sensitivity and resolution. Although there are several corresponding reports, it is rarely used in practical applications. On this basis, we combine the contact input electrode and contactless output electrode boldly and design a new conductivity detection device, which can ensure the response sensitivity of the test object and avoid problems caused by contact electrode as much as possible. It has great application potential in miniaturization and large-scale industrial analysis and detection. To make it more realistic and accurate for the experimental data, we used ultrapure water with low conductivity as the solvent.

## 2. Materials and Methods

### 2.1. Materials and Instrumentation

The sample solution was prepared with potassium chloride purchased from Shanghai Titan Scientific Co., Ltd. (Shanghai, China). Ultrapure water (18.2 MΩ∙cm water) was obtained from a Millipore water purification unit. The test pool device consists of a straw with a diameter of 0.6 cm and a length of 18 cm as the channel, two silver wires with a diameter of 0.3 mm perforated as the input electrode, copper foil as the output electrode, and silicone rubber to seal the pores. The testing pool is fixed in the shielding box. Signal input is provided by a waveform generator (DG4102, RIGOL Technologies Inc., Jiangsu, China), with a sinusoidal signal of up to 20 V_PP_. Dc output was measured using the module AD637 (Shenzhen Yanglin Electronics Co., Ltd., Shenzhen, China) and Chromatography Data System N2000 (Zhejiang University Information Engineering Co., Ltd., Zhejiang, China). Digital oscilloscope (DS1102E, RIGOL Technologies Inc., Jiangsu, China) detects signal strength and waveform changes during the process. In order to reduce the interference of background signal and expand the gain, the module AD637 was replaced by a DSP lock-in amplifier (OE1022, Guangzhou Saienkeyi Electronic Technology Co., Ltd., Guangzhou, China). A peristaltic pump (Baoding Qili Constant current pump Co., Ltd., Baoding, China) is equipped with appropriate flow rate to drive the sample for injection.

### 2.2. Device Connection and Measurement Setup

As shown in Figure 1a, the sensor device of the detector is operated by connecting the signal unit, detection pool, amplifier, and display to measure the concentration of potassium chloride ions. Figure 1b shows the diagram of the sensing section of DISODCD. Ultrapure water was used to prepare potassium chloride solutions with various concentration gradients. During configuration, argon was passed into a beaker and a volumetric flask in advance to discharge the air inside so as to avoid CO_2_ dissolving into ultrapure water and to form carbonate ions, which increased the conductivity of the solution and thus deviated from the actual value. Samples should be tested in a relatively sealed environment and injected quickly to avoid CO_2_ dissolving in the air.

### 2.3. The Theoretical Electronic Circuit Section of Capacitive Sensor

Different from the traditional capacitance sensor, the capacitance sensor neutralizes the advantages and disadvantages of the detection cell circuit. The structural schematic diagram of DIC^4^D proposed by our research group in previous work [21] is shown in Figure 2. Where C_L_ is the capacitance of wall, R_x_ is the equivalent resistance of the solution, and C_S_ is the stray capacity. For capacitors generated in a circuit, the value of C_L_ is usually large, and the value of C_S_ is small.

The structural design of the sensor is shown in the Figure 3. In Figure 3a, R_L_ is the equivalent resistance of electrode lead, C_1_ is the double layer polarization capacitance caused by electrode polarization, R_1_ is the equivalent polarization resistance formed by electrode polarization effect, R_x_ is the solution resistance, C_P_ is the equivalent capacitance of electrode lead distribution, C_0_ is the capacitance of wall, and A is the operational amplifier.

In general, R_L_ is negligible. Since C_1_ provides a path for the AC signal, R_1_ is short-circuited by C_1_. The value of C_1_ is usually very large, and the value of C_p_ is very small. According to X_C_ = 1/ωC, the value of X_C1_ is very small, and the value of X_Cp_ is very large. When the high purity solution is measured, the solution impedance R_x_ is high, the capacitive reactance X_C1_ is small and can be ignored, while the effect of C_p_ cannot be ignored. On the contrary, when the high concentration solution is measured, the solution impedance R_x_ is small, C_p_ can be ignored, while the effect of capacitive reactance X_C1_ cannot be ignored. The equivalent circuit diagram of Figure 3a is simplified to Figure 3b, where C_0_ is the capacitance sum of C_W_.

According to the electrical model in Figure 3c, Z_1_ can be expressed as
(1)$$Z_{1}=\frac{1}{j\omega C_{1}}+\frac{1}{j\omega C_{w}}+R_{x}$$

Z_1_ is equal to Z_2_ in the stable state after the flow passes through. I_3_ is the current flowing out of the connection point, and Z_3_ is the load impedance formed by the detection circuit. When the section of the load circuit is fixed, the value of Z_3_ is fixed, and we regard it as a constant for calculation.

According to the superposition theorem, the current of electrode lead distribution Ĩ_P_ can be expressed as
(2)$${\tilde{I}}_{P}={\tilde{U}}_{1}\cdot j\omega C_{P}-{\tilde{U}}_{2}\cdot j\omega C_{P}$$

When the input signals are identical, which means Ũ_1_ is equal to Ũ_2_, and I_P_ is equal to 0, equivalent to the open circuit state, any concentration of the solution is not affected by C_p_.

According to Kirchhoff’s law, |Ĩ_3_Z_3_| is
(3)$$S=\left|{\tilde{I}}_{3}Z_{3}\right|=\left|\frac{{\tilde{U}}_{1}Z_{2}Z_{3}+{\tilde{U}}_{2}Z_{1}Z_{3}}{Z_{2}Z_{3}+Z_{1}Z_{3}+Z_{1}Z_{2}}\right|$$

The impedance in the branch is equal to Z_A_ when there is only background electrolyte in the test cell. The impedance in the branch is equal to Z_B_ when there is only analyte in the test cell. In the case of the input AC signal is exactly the same, the formula for S can be simplified to
(4)$$S=\left|{\tilde{I}}_{3A}Z_{3}\right|=\left|\frac{2{\tilde{U}Z}_{3}}{2Z_{3}+Z_{A}}\right|$$

The difference ∆S of the response signal between the concentration of analyte and the concentration of background ultrapure water is
(5)$$\Delta S=\left|{\tilde{I}}_{3B}Z_{3}\right|-\left|{\tilde{I}}_{3A}Z_{3}\right|=\left|\frac{2{\tilde{U}\;Z}_{3}}{2Z_{3}+Z_{B}}-\frac{2{\tilde{U}\;Z}_{3}}{2Z_{3}+Z_{A}}\right|$$

According to the expression of S, the ratio of analyte to background ultrapure water signal, P/N, was given by
(6)$$P/N=\frac{\left|{\tilde{I}}_{3B}Z_{3}\right|}{\left|{\tilde{I}}_{3A}Z_{3}\right|}=\frac{\left|2Z_{3}+Z_{A}\right|}{\left|2Z_{3}+Z_{B}\right|}$$

The value of C_w_ is half of the decomposition of the wall capacitance C_L_ with the same parameters of input and output electrodes, which can lead to C_w_ < C_L_. The C_1_ generated for the contact electrode is very large [34]. C_s_ is a capacitor of air as the medium, and its impedance is much larger than that of other capacitors, so the capacitance is very small and can be almost ignored. Hence
(7)$$C_{s}\ll C_{w}\ll C_{L}\ll C_{1}\;$$
(8)$$Z_{s}\gg Z_{w}\gg Z_{L}\gg Z_{1}\;$$
(9)$$Z_{1\left(\mathrm{DISODCD}\right)}<R_{x}<Z_{\left({\mathrm{DIC}}^{4}D\right)}$$

According to Equations (4)–(6), the following equations were obtained
(10)$$S_{\mathrm{DISODCD}}-S_{{\mathrm{DIC}}^{4}D}>0\;$$
(11)$${\Delta S}_{\mathrm{DISODCD}}-{\Delta S}_{{\mathrm{DIC}}^{4}D}>0$$
(12)$$P/N_{\mathrm{DISODCD}}-P/N_{{\mathrm{DIC}}^{4}D}>0$$

After derivation, we can find that the detection signal of DISODCD should be much larger than that of DIC^4^D and C^4^D in theory, which is proved by Equation (10). Although we have not performed theoretical derivation for C^4^D, it has been discussed in detail in Wang’s article [35]. Equation (11) shows that when the concentration in solution changes from A to B, the difference value of DISODCD of signal response is large, which means that the resolution of low concentration is improved. From Equation (12), although the response signal of background ultrapure water increased, the ratio of analyte to its signal did not decrease but increased instead. Our improved device can not completely eliminate the reactance effect caused by electrode polarization to some extent, but this effect can be reduced to some extent on the premise of improving performance.

## 3. Results

### 3.1. Optimization of Conditions

In the experiment, we adopted the maximum peak voltage of 20 V_PP_ and the initial parameter conditions of 0 degrees with the same phase and adopted the control variable method to select the appropriate measurement conditions by changing the parameters such as frequency, voltage, and phase, combining the ratio of the signal and low concentration resolution to obtain the best test conditions. Figure 4 shows that the frequency at 40 kHz is the optimal value (Figure 4a), and the higher the peak voltage is, the stronger the received signal (Figure 4b) will be. The difference of received signal corresponding to low concentration is the largest when 0° is in phase (Figure 4c), and the ratio of the signals is also the largest (Appendix A). In contrast, when 180° is in phase, the difference is opposite. In addition to the parameter conditions of the signal source, the effects of the input electrode spacing and the width of the coupled output electrode on the ratio of the signal are also studied. Due to the low resolution of low-concentration samples, in order to be clearer and more intuitive to see the comparison, we use 0.01 mM potassium chloride solution.

In particular, the optimization of experimental conditions is carried out on module AD637. The lock-in amplifier has the requirements of the range, so the optimal conditions under various parameters cannot be observed completely and intuitively. After replacing the lock-in amplifier, we adjusted the subsequent experimental conditions to meet the optimal conditions of the range. In the literature, the best waveform is usually the signal wave of a sine wave or pulse wave. This paper also makes a simple exploration on the selection of waveform; the results of the ratio of the signal are shown in Table 1. According to the test results in the table, we used the sinusoidal waveform as the signal output waveform in subsequent experiments. When the electrode spacing is 2 mm, and the coupling output electrode width is 10 mm, the optimal ratio of the signal is achieved. Smaller distance between the electrodes brings lower impedance of the solution and greater signal strength. Wider distance between output electrodes results in smaller partial voltage on the coupling capacitance and larger output voltage value, but the ratio of the signal does increase. When the electrode width reaches an extreme value, the partial pressure does not change. From the perspective of material saving, we choose the width of 10 mm.

### 3.2. Shielding Performance Tests

The sensor detection pool is placed in the shielding box, and the anti-interference performance of the device is verified by applying an external signal interference on the solution pipe outside the shielding box. As shown in Figure 5, when the sample in the cell is ultrapure water, signal interference occurs in the shielding box before 3 min and after 8 min, and the response signal is basically unchanged with good reproducibility after repeating the test twice. On the contrary, the response value varies irregularly with the external signal interference in the case of no shield for about 3–8 min. It indicates that the device has strong ability to resist external interference. In order to show the effect of the shielding box on noise interference to the greatest extent, we chose module AD637 to replace the lock-in amplifier for detection because the phase lock reduced the interference signal to a certain extent.

### 3.3. Response to the Test Process and Results

Figure 6 shows the monitoring diagram of the detection process of each sensing part of the sensor visually. It can be seen from Figure 6a,b that the output signal of the larger excited sinusoidal wave is significantly weakened after the voltage signal passes through the coupled capacitive reactance in the circuit. It can also be explained that the excitation signal can propagate relatively intact through the contact electrode because the capacitance generated by the electrode polarization of the contact electrode is large enough and the capacitance reactance is small. Both the solution impedance and the coupling capacitance of the non-contact electrode produce high reactance, which results in a greatly reduced voltage signal from the detection cell.

In past work, our group proved that DIC^4^D has better performance than C^4^D [21]. Here, we compare the improved sensor DISODCD with DIC^4^D and DCD to detect the same potassium chloride sample (Figure 7). Due to the limited range of the lock-in amplifier, we use the AD637 module to collect data in order to observe the process of peak change. We found that the signal strength of DCD is higher, but the resolution at low concentration is lower, which may be due to the close distance between the contact electrodes, resulting in the large stray capacitance and the equivalent capacitance of electrode lead distribution C_P_, so that the low concentration has a great influence and cannot be distinguished. DIC^4^D has a lower signal and resolution, which is proved by Equations (10) and (11). Under the same experimental conditions, at S/N = 3 or according to the formula LOD = 3 δ/S, the LOD of DISODCD for potassium chloride solution is calculated as 1 nM.

In order to further observe the response of a certain concentration range, we test under-phase-locked conditions. Since the range exceeds the system test range, we use a millivoltmeter to collect data at the peak point. In Figure 8a, the detection range of DISODCD sensor is 0.01 μM to 10 mM in actual testing for a single sample. DCD has a very high response value at high concentrations. DIC^4^D and DISODCD have better resolution at low concentrations, but the response of DIC^4^D is low, and the concentration range that can respond is smaller. Figure 8b reflects the ratio of 0.01–10 μM KCl solution to ultrapure water. At low concentrations, the response ratio of DISODCD is greater than that of DCD and DIC^4^D. Based on the response value in Figure 8a, we calculated the ratio of 10 μM KCl solution to ultrapure water and found that the ratios of DISODCD, DIC^4^D, and DCD were 1.27, 1.19, and 1.20, respectively, which shows that the sensitivity and resolution of DISODCD are higher than the other two sensors in the low concentration range.

### 3.4. The Ability to Resist Interference from Impurities

With the increase of the number of sensor tests, especially the viscous or adsorbed sample solution, the detection pool is greatly polluted and difficult to clean. In order to explore the interference of impurities on the sensor sample, we first injected the vegetable oil and made it adhere to the test pool. We rinsed it with ultrapure water several times and then passed the sample KCl solution into the flow cell. As shown in Figure 9a, the three types of sensors, DIC^4^D, DISODCD, and DCD, all have slight deviations from the normal value. After three tests, the deviated signal value gradually approaches the normal value. Because the technical parameters of the lock-in amplifier have a certain range, we use module AD637 to collect the same response value in the concentration range of 0.1 to 10 mM and observed that the deviation value of DCD was greater than DIC^4^D and DISODCD (Figure 9b), and this effect might be more obvious when amplifying the picked response signal or enhancing the excitation signal. Impurity interference has little effect on the signal response of the three sensors. The impact on DIC^4^D and DISODCD is less than that of DCD. For DIC^4^D, the reason why impurity pollution has little effect on the response is that some impurities are attached to the tube wall, and the change in the properties of the two plates of the equivalent capacitance affects the capacitance of the electrode. But the electrode itself is not affected, that is, its ability to conduct current is not affected. Therefore, the impact of impurity pollution on DIC^4^D is relatively small. The layer of impurities attached to the contact electrode is equivalent to a layer of dielectric that is wrapped on the outside unevenly, forming a capacitive conduction effect, which has nothing to do with impurities on the tube wall, although, it affects the ability of the electrode to conduct current. The contact electrode of DISODCD is less than the DCD electrode in number, and the influence on the electrodes is also small. Furthermore, the contamination of this non-stable attached electrode also varies greatly in response over time, which is the result of the irregular movement of impurities in the detection cell (Appendix A). Regarding the experiment of impurity contamination, we measured it under the condition of 40 min after injections. At this time, the detection cell is in a relatively stable state. For the first 40 min of injections, we capture data every 5 min because the response data caused by the shedding and dissociation of pollutants changes greatly and have no reference value.

### 3.5. Reproducibility Verification

Considering the interference of accidental factors, we verified the repeatability of the detector test results. From Figure 10, intuitively, we can see that after 10 consecutive tests, the response value of 10 nM potassium chloride solution fluctuates around the voltage line of 137.5 mV at each gradient point. The sensor is accompanied by common electronic noise. The effect of noise on DISODCD is weak, but there is a low baseline drift, which makes the whole baseline fluctuate around the voltage line of 136.5 mV. The difference in shape, width, and peak point across the response is small, which shows that the detector has good reproducibility and can be used as a general purpose sensor. In the measurement process, we have to ensure that the repeated tests are carried out in the same environment and the experiments are under optimal conditions. In addition, it is worth noting that when verifying low concentration solution, the fresh solution is selected to avoid air entering the liquid to interfere with real results.

## 4. Conclusions

In this work, a new type of conductance detection sensor, DISODCD, is constructed. On the basis of improved sensitivity and resolution, the parallel circuit cancels part of the adverse capacitance reactance introduced by electrode polarization and reduces the effect of the impedance caused by the coupling wall capacitance on the measured solution resistance. The detection range is 0.01 μM to 10 mM in actual testing, and the LOD is 1 nM. The ratio of the response of 10 μM KCl solution to ultrapure water tested by the sensor is 1.27, which is better than DCD and DIC^4^D. The experimental results also show that the impurity contamination has little effect on the sensor. This work provides a new idea for improving the performance of conductivity measurement of the sensor and a feasible solution for conductivity measurement of organic and inorganic ions.

## Figures and Tables

**Figure 1 sensors-22-02729-f001:**
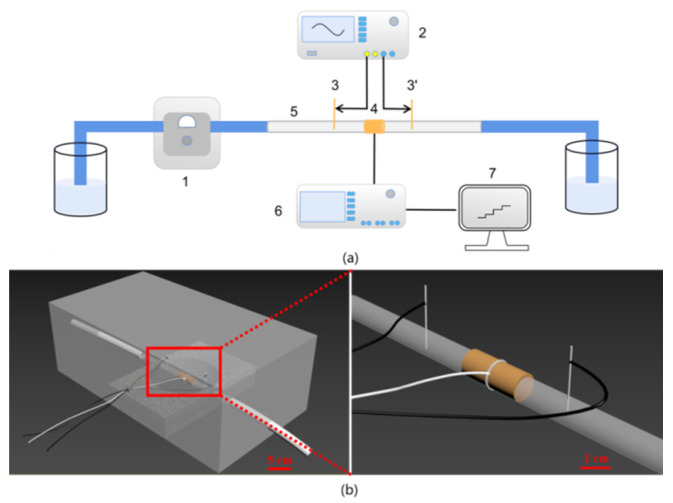
The measurement principle of DISODCD. (**a**) A plan of the experimental setup: (1) Peristaltic pump; (2) Function generator; (3) Input electrodes; (4) Output electrode; (5) Plastic pipe; (6) Lock-in amplifier; (7) Signal collector. (**b**) The model figure of the sensing section.

**Figure 2 sensors-22-02729-f002:**
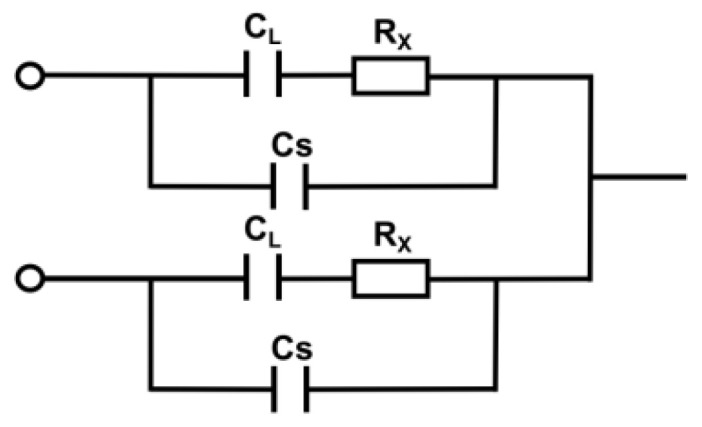
The equivalent circuit model of DIC^4^D.

**Figure 3 sensors-22-02729-f003:**
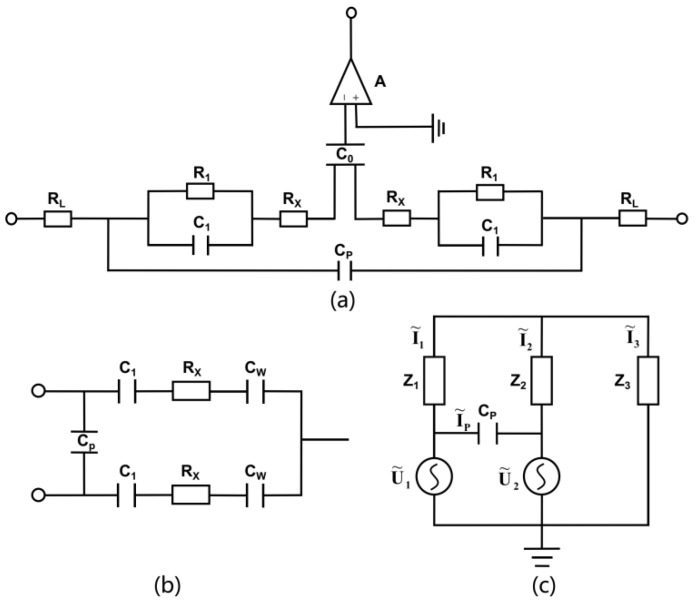
The equivalent circuit models of DISODCD. (**a**) Original equivalent circuit of DISODCD; (**b**) Simplified equivalent circuit of DISODCD; (**c**) The equivalent circuit of DISODCD used by the superposition theorem.

**Figure 4 sensors-22-02729-f004:**
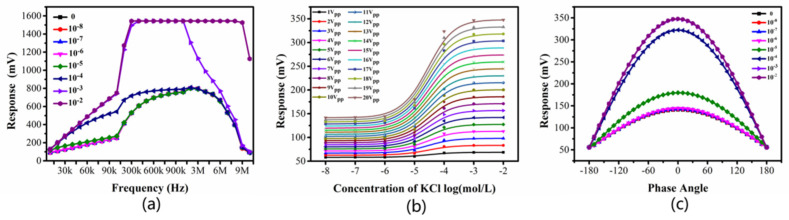
Experimental results of parameter optimization. (**a**) Frequency; (**b**) Concentration of KCI log; (**c**) Phase angle.

**Figure 5 sensors-22-02729-f005:**
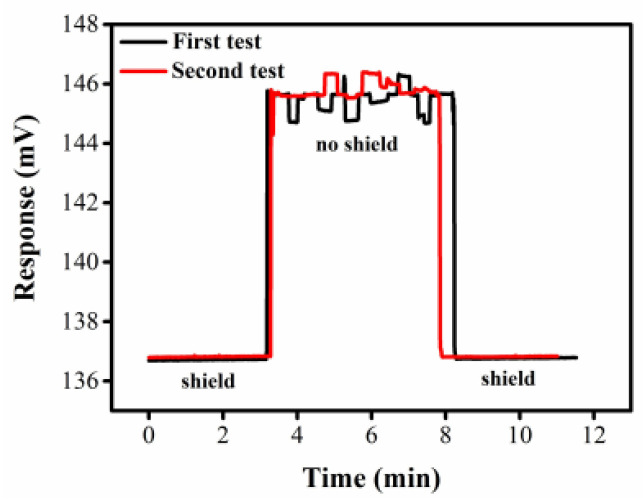
Output signals of the ultrapure water by DISODCD sensor with or without shield configuration.

**Figure 6 sensors-22-02729-f006:**
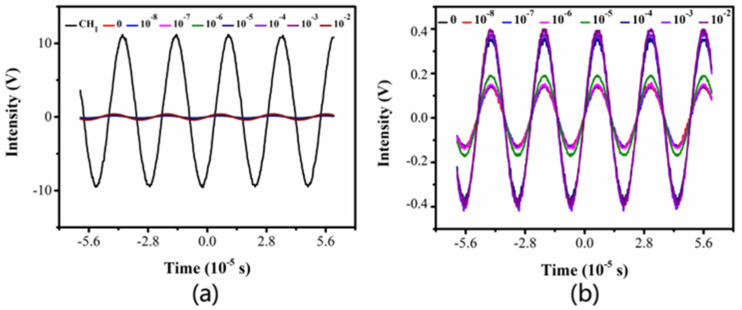
Signal waveform of 0–10 mM KCl solution at the input and output ends of the detection cell. (**a**) The input and output ends; (**b**) The output end.

**Figure 7 sensors-22-02729-f007:**
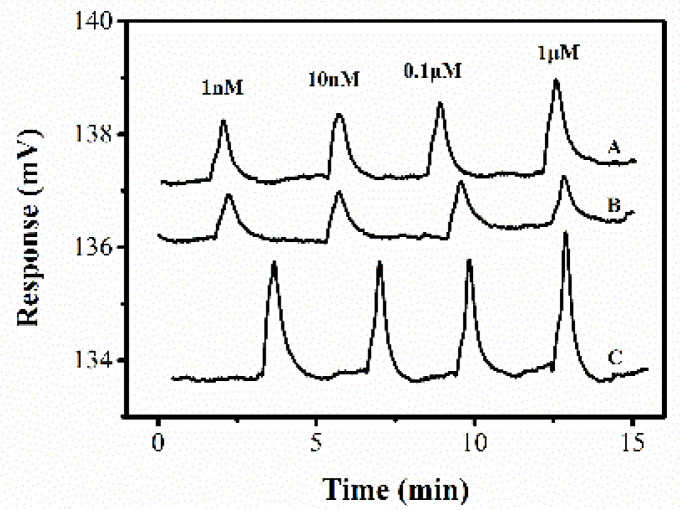
Response of DISODCD (A), DCD (B), and DIC^4^D (C) to 0.001–1 μM KCl solution.

**Figure 8 sensors-22-02729-f008:**
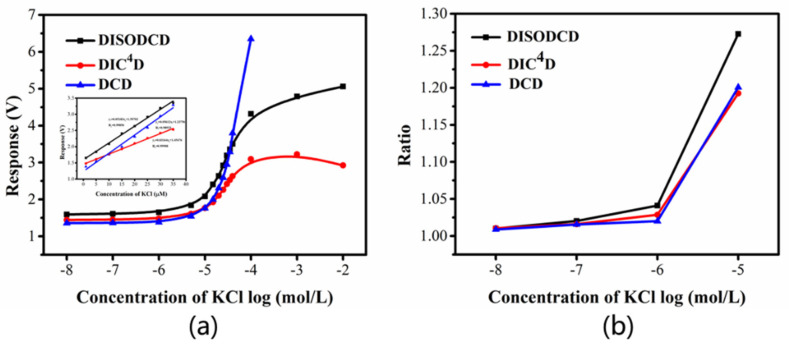
Response of DISODCD, DIC^4^D, and DCD to 0.01 μM-10 mM KCl solution. (**a**) Response curve of DISODCD, DIC^4^D, and DCD. (**b**) The ratio of the response of DISODCD, DIC^4^D, and DCD.

**Figure 9 sensors-22-02729-f009:**
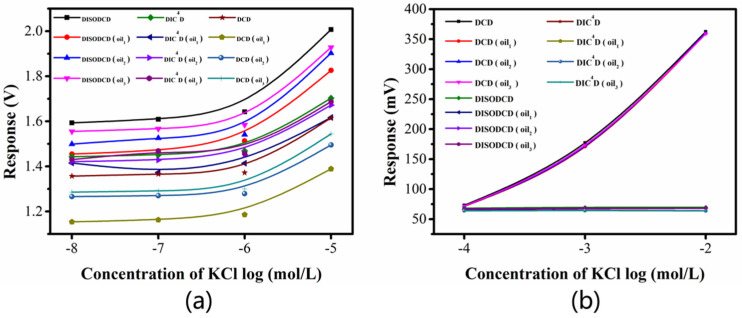
Signal response diagram of DCD, DIC^4^D, and DISODCD sensors to KCl solution disturbed by oil impurities. (**a**) Response graphs of 0.01–10 μM KCl solution were tested 3 times; (**b**) Response graphs of 0.1–10 mM KCl solution were tested 3 times.

**Figure 10 sensors-22-02729-f010:**
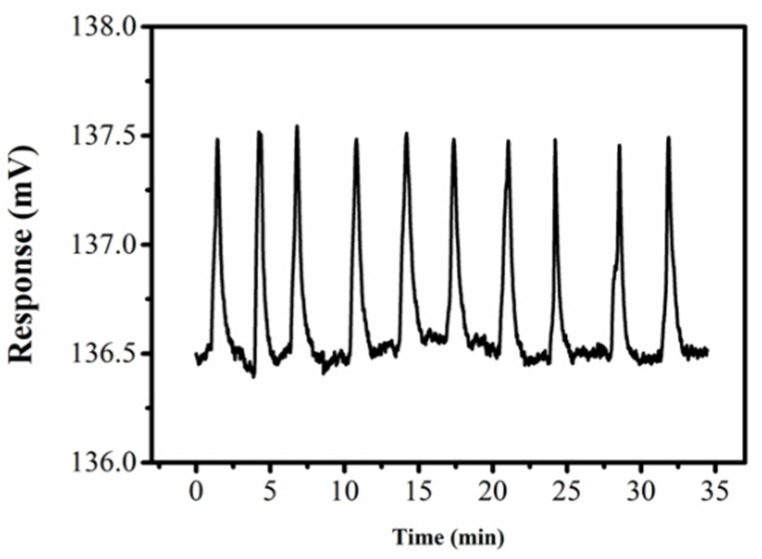
Response curve of DISODCD for 10 repeats to 10 nM KCl solution.

**Table 1 sensors-22-02729-t001:** The result of researching the ratio of the output waveform.

Parameter		Ratio
Length of gap (mm)	10	1.08
	8	1.09
	5	1.11
	2	1.14
Length of electrode (mm)	5	1.10
	8	1.11
	10	1.14
	12	1.13
	15	1.12
Wave form	Sine	1.14
	Square	1.07
	Ramp	1.11
	Pulse	1.12

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
