# Peer review of "A Study on Double Inputs Direct Contact and Single Output Capacitively Coupled Conductivity Detector"

_sensors, 2022, doi:10.3390/s22072729_

Round 1

Reviewer 1 Report

Comments are included in an attached file.

Reviewer 2 Report

This manuscript by Zhang et al. presents a new direct contact conductivity detection method with high sensitivity and low detection limit which is of interest to many basic chemical analysis applications. This manuscript is recommended to be accepted after revision. I have a few questions and comments the authors need to address.

  1. The title case is problematic. Either use all capitalized words or use the sentence case.
  2. A simplified schematic of the experimental setup should be positioned and discussed before or together with the equivalent circuit after which the comprehensive schematic can be shown. The depiction of the circuit from their previous is out of touch with the flow of the text and can be moved to appendix. Fig. 3 also need scale bars. 
  3. What is the physical meaning of the ratio signal in the paper? Is it proportional/related to the solution concentration ratio directly? 
  4. The authors need to justify the clams about the value relationships about Cs and Rs in section 2.1. What are the actual typical quantities of the parameters in Eq. 7-9? 
  5. The derivation of Eq 10-12 is not clear. What is the value of Z3 which appears in Eq. 4-6 compared to other impedances to lead to Eq. 10-12? 
  6. In section 3.1, why is the frequency of 40kHz the optimal? How do they conclude from Fig. 4c that "When 0° is in phase, the difference of the received signal corresponding to the low concentration is the largest but the ratio of the signal is the smallest. In contrast, when 180° is in phase, the difference is opposite"? It is hard to see without a direct plot of the ratio data. 
  7. What material and structure do they use for the shielding box? 
  8. I cannot see clearly how the impact of impurity pollution on DIC4D and DISODCD is relatively small. A plot for the signal deviation from the uncontaminated measurements for each method is needed. In addition, can the authors confirm that the same test pool is used for all conductivity detectors? The data for DCD oil3 is a single line with no experimental dots. Is there real data for it? 

Reviewer 3 Report

The authors are missing many related articles of previous works in a similar area using more advanced techniques and/or experimental setups.  The method discussed has been done before, so the researchers must show how their work advances the area.  This is really the major deficiency on how to differentiate and show how this advances techniques used for decades. 

The writing is well-done, with only some small errors.  All units should have a a space between the number and unit.  Also, some other small errors but these can be fixed easily.

The graphs are difficult to read sometimes, and the quality should be increased so that people can read all writing and axes titles. 

Previous versions of this sensors should be better defined, so that the improvements and differences in the performance can be related to the changes.  These other designs are included in graphs, but not really defined (only referenced which other would have to go find those papers and compare).  The responses should be really compared to the response of other electrode designs discussed in other works, to show the advantages of this work. 

Statistical analysis for the data is missing to show variation of the response in different concentrations and contaminated solutions (and over time).  These experiments are relatively easy to run, so more statistics should be included in the data.

Round 2

Reviewer 2 Report

I thank the authors for addressing my comments and questions. This manuscript can be published now. 

Author Response

Thank you very much for your time involved in reviewing the manuscript and your careful and thoughtful comments on our manuscript. These suggestions have important guiding significance for my writing and research work.

Reviewer 3 Report

The authors addressed my three of my four major concerns from the previous version of this paper.

1) The first concern was related to stressing the originality of this work.  These methods have been generally completed before for many systems and electrode configurations, but the further table does help to differentiate your modifications and the effect on contamination and level of detection that is improved with these changes.

2) The description of how the experiments were completed was added to the paper so anyone can repeat.  This is very important, and this addition is appreciated.

3) The graphs were very difficult to read, where the headings were very small and not distinguishable.  Now it is possible to read them due to the change in these figures by the authors.

4) The statistical addition was not completely added to this version.  Yes, an additional supplementary figure was added, which shows results from a few different concentrations over time.  This is fine.  What was really meant was that if this experiment was completed multiple times, did the researcher see a variance in the results (i.e. what is the spread in data, even expressed simply by a standard deviation from multiple measurements).  This system is not hard to setup and run, so the authors could of showed the spread in data versus time (versus concentrations).   If this data is not available, then can the authors at least place a statement in the paper with how many samples were run for the graphs, and if the data represents the general trend of multiple runs.   
